# Estimating class separability of text embeddings with persistent homology

**Kostis Gourgoulias**                                                              *kgourgou@pm.me*
*JPMorgan Chase & Co.*

**Najah Ghalyan**                                                        *najah.ghalyan@jpmchase.com*
*JPMorgan Chase & Co.*

**Maxime Labonne**
*JPMorgan Chase & Co.*

**Yash Satsangi**
*JPMorgan Chase & Co.*

**Sean Moran**
*JPMorgan Chase & Co.*

**Joseph Sabelja**
*JPMorgan Chase & Co.*

**Reviewed on OpenReview:** *https://openreview.net/forum?id=8DWrIMuLya*

## Abstract

This paper introduces an unsupervised method to estimate the class separability of text datasets from a topological point of view. Using persistent homology, we demonstrate how tracking the evolution of embedding manifolds during training can inform about class separability. More specifically, we show how this technique can be applied to detect when the training process stops improving the separability of the embeddings. Our results, validated across binary and multi-class text classification tasks, show that the proposed method's estimates of class separability align with those obtained from supervised methods. This approach offers a novel perspective on monitoring and improving the fine-tuning of sentence transformers for classification tasks, particularly in scenarios where labeled data is scarce. We also discuss how tracking these quantities can provide additional insights into the properties of the trained classifier.

## 1 Introduction

Pre-trained language models, having undergone extensive preliminary training, have demonstrated remarkable versatility. They have shown the capacity to generalize to new tasks from a mere handful of examples by employing techniques such as in-context few-shot learning (Brown et al., 2020), parameter-efficient fine-tuning (PEFT) (Liu et al., 2022), and pattern-exploiting training (Schick and Schütze, 2020a). Commonly, these methodologies necessitate the use of language models that vary in size from a few hundred million (Lan et al., 2019; Schick and Schütze, 2020b) to billions of parameters, exemplified by GPT-3 (Brown et al., 2020) and T0 (Sanh et al., 2021). Furthermore, these techniques often involve the conversion of a task, such as classification, into a language modeling format, akin to a cloze question.

Text embedding is a crucial part of any language model that maps text to numerical feature vectors that can be fed to downstream machine learning operations aimed at performing specific tasks, e.g., text classification. Texts with similar meanings need to be mapped to feature vectors closely spaced in the embedding space,

while texts with largely distinct meanings need to be mapped further away from each other. If the pre-trained language model is used for text classification, metrics capturing class separability can be used to assess how good or bad a text embedding is performing. In this case, a good embedding process would generate an embedding manifold with high class separability so that a downstream classification process would perform well.

In classification problems, class separability metrics capture how easily features distinguish their corresponding classes (Fukunaga, 2013). The intuition for adopting these metrics for feature ranking is that we expect good features to embed objects of the same class close to each other in the feature space, while objects of different classes are embedded far away from each other. This can measure the discriminative power of a feature (Rajoub, 2020). Having those metrics can also assist with capturing the minimum complexity of a decision function for a specific problem, e.g., VC-dimension (Vapnik, 1998). Typically, separability metrics range from model-specific, also known as "wrapper" methods (e.g., Gini importance), to more generic "filter" methods that capture intrinsic properties of the data independently of a classifier (e.g., mutual information). A simple such metric is Fisher's discriminant ratio, which quantifies the linear separability of data by using the mean and standard deviation of each class (Li and Wang, 2014). However, this metric comes with strong assumptions of normality. Geometrically-inspired methods (Greene, 2001; Zighed et al., 2002; Guan and Loew, 2022; Gilad-Bachrach et al., 2004) look at the distances and neighborhoods of features to give a model-independent view of how well-separated the classes are. However, high-dimensional settings and/or large datasets can be challenging for most separability metrics. Most information-theoretic estimators do not scale well with data size or require training separate models (Belghazi et al., 2018). Geometric methods are attractive as they are not tied to a classifier, but can be informative only for specific separation regimes (Mthembu and Marwala, 2008), and are impacted by the complexity of computing graph neighborhoods and distances between the points. Moreover, a common requirement for these class-separability metrics is that they require labeled data, making them mainly limited to the supervised learning setting.

In mathematics, homology is a general way of associating a sequence of algebraic objects, such as abelian groups, with other mathematical objects, such as topological spaces, e.g., data manifolds (Hatcher, 2002). The fundamental groups of topological spaces, introduced by Poincaré (Munkres, 2018), are the first and simplest homotopy groups (Hatcher, 2002) and are algebraic invariants that are critically important for characterizing and classifying topological spaces (Massey, 1991).

Topological data analysis (TDA) is a mathematical framework that uses topological concepts to analyze and understand complex data sets. TDA is increasingly being used in machine learning to extract information from high-dimensional data that is insensitive to the choice of metric, allowing for more robust analysis. TDA also provides dimensionality reduction and robustness to noise (Carlsson, 2009). One of the most important techniques in TDA is persistent homology (PH) (Malott and Wilsey, 2019), which analyzes the topological features of a data set across different scales. Persistent homology tracks the birth and death of topological features, such as connected components, loops, voids, and identifies those that persist as the filtration threshold increases Lazar and Ryu (2021). This approach provides a more nuanced understanding of the underlying structure of the data and allows for clustering and data analysis.

In this context, Barannikov et al. (2022) introduced a topologically-inspired approach for comparison of neural network representations by introducing R-Cross-Barcode tool, which measures the differences in the multi-scale topology of two point clouds. Based on this tool, they proposed a Representation Topology Divergence (RTD) that can measure multi-scale topological dissimilarity between two representations. In parallel, Garrido et al. (2023) introduced a method — coined RankMe — that allows for assessing the performance of Joint-Embedding Self Supervised Learning (JE-SSL) methods providing informative data representations, even on different downstream datasets, without requiring any labels. Their experiments demonstrate RankMe's ability to inform about JE-SSL downstream performances across different methods, e.g. VICReg, SimCLR, DINO, and their variants, and across architectures, e.g. using a projector network and/or a nonlinear evaluation method. Furthermore, results show that RankMe can enable hyperparameters cross-validation for JE-SSL methods, and is able to retrieve and sometimes surpass most of the performance previously found by manual and/or label-guided search while not employing any labels, on both in-domain and out-of-domain datasets. In the context of persistent homology for characterizing text data, Tulchinskii et al. (2023) developed an algorithm for estimating the intrinsic dimensionality of natural language text

that can be used to distinguish between human-generated and AI-generated text. Their experiments show that the intrinsic dimension can serve as a good score for artificial text detection with reduced bias against non-native speakers.

## 1.1 Related Work

Hajij and Istvan (2021) framed the classification problem in machine learning by expressing it in topological terms. Using this topological framework, they showed the circumstances under which the classification problem is achievable in the context of neural networks. While we do not make direct use of this formalism, some of our experiments are motivated by the discussion in this work.

Rieck et al. (2019) developed a complexity measure for deep neural networks, called "neural persistence", using algebraic topology. This measure was used as a stopping criterion that shortens the training process while achieving comparable accuracies as early stopping based on validation loss by taking into account the layers and weights of the whole model. Pérez-Fernández et al. (2021) represented neural networks as abstract simplicial complex, analyzing them using their topological fingerprints via persistent homology. They then described a PH-based representation proposed for characterizing and measuring the similarity of neural networks. Experiments demonstrated the effectiveness of this representation as a descriptor of different architectures in several datasets. While there are similarities with our work, we do not use persistent homology to compare different models. Instead, we explicitly focus on examples from classification and only use information from the first homology group of the text embeddings, $H_0$, that we get from our model to assess separability. This homology group captures information about the connected components in the data.

Gutiérrez-Fandiño et al. (2021) suggested studying the training of neural networks with Algebraic Topology, specifically persistent homology. Using simplicial complex representations of neural networks, they studied the persistent homology diagram distance evolution on the neural network learning process with different architectures and several datasets. Results showed that the persistent homology diagram distance between consecutive neural network states correlates with the validation accuracy, implying that the generalization error of a neural network could be intrinsically estimated without any holdout set. While we are also interested in getting some signal about validation performance, our approach uses only the simplest topological information from the embeddings, $H_0$, instead of explicitly taking into account the weights and connections of the whole model. We also track not just statistics of death times, but also their overall density as epochs progress.

Griffin et al. (2023) showed that the topological structure of training data can have a dramatic effect on the ability of a deep neural network (DNN) classifier to learn to classify data. Previously, Naitzat et al. (2020) highlighted that DNNs tend to simplify the topology of input as it gets passed through the DNN's layers. Both of those works are connected to ours through the tracking of changes in the topology of embeddings during training and how that affects the performance of the DNN.

## 1.2 Contributions

Primary contributions of the paper are summarized below:

- *An unsupervised method for class-separability estimation*: We use information from the 0-homology groups of data manifolds to understand the class-separability of embeddings during model training. Unlike standard supervised techniques, which require labels for computing class separability, the proposed method can estimate class separability without them. In addition, in contrast to metrics that capture clusters and the distance between them, like the Calinski-Harabasz (CH) Index, we present evidence that this topological view of separability aligns more in terms of behavior with the supervised metrics.

- *Experimental analysis of sentence-transformer embeddings on $H_0(X)$ density space*: The paper involves experimental analysis of embedding manifold evolution over training epochs based on the densities of $H_0(X)$ persistence times. By tracking those densities, we see how the embedding space

is organized during training. We discover that normalizing the embedding layer causes the embeddings to organize in well-defined connected components during fine-tuning.

- *Comparison of the behavior of different measures of separability on high-dimensional spaces*: As part of our experiments, we compare the behavior of various metrics of separability when applied to high-dimensional embedding spaces. We find that supervised metrics of separability have similar behavior during training, whereas some unsupervised metrics can miss important details like the diminishing return of further training. In the experiments, we also show how the proposed (unsupervised) metric does not have this issue.

### 1.3 Organization

The paper is organized into six sections, including the present section. Section 2 provides a brief introduction to homology groups and persistent homology of data manifolds. This background is essential for understanding the proposed method for estimating class separability of datasets presented in Section 3 along with the baseline metrics in Section 3.1. Two sets of experiments are presented in Section 4 covering binary and multi-class cases. Section 5 covers computational costs of the study as well as a discussion on the proposed statistic and the behavior of the persistence times during model training. Section 6 summarizes and concludes the paper along with recommendations for future research.

## 2 Background on Persistent Homology

We now briefly introduce the essential notation and concepts of persistent homology that we will use in this work. For more details, please see the supplementary material or Section 2 from Naitzat et al. (2020).

Suppose we have a collection of points $X = \{x_1 \ldots, x_N\} \subset \mathbb{R}^d$ and a norm $\|.\| : \mathbb{R}^d \times \mathbb{R}^d \to \mathbb{R}$. At a high level, persistent homology (PH) concerns itself with identifying the shape and topological features of data manifolds in a way that is robust to noise. To be able to identify those, we can start with a standard construction in PH, the Vietoris-Rips[1] set at scale $\epsilon$:

$$\mathrm{VR}_\epsilon(X) := \{\sigma \subset X : \sigma \neq \emptyset, \forall x, y \in \sigma, \|x - y\| \leq \epsilon\}. \tag{1}$$

For two elements to belong to the same $\sigma$, their $\epsilon$-balls need to intersect. Thus $\mathrm{VR}_\epsilon(X)$ generates a filtration, called the Vietoris-Rips filtration, on the normed space $(X, \|.\|)$, where $\mathrm{VR}_0(X) = \{\{x\} : x \in X\}$ and $\mathrm{VR}_\infty(X) = \{\{x_1, \ldots, x_N\}\}$.

In PH, the Vietoris-Rips set is called an *abstract simplicial complex*. If we interpret every $\sigma \in \mathrm{VR}_\epsilon(X)$ as describing a relationship between the points $x \in \sigma$, we can construct a geometric realization of $\mathrm{VR}_\epsilon(X)$ by building a graph with vertices the points in $X$ and edges described by the relationships in $\mathrm{VR}_\epsilon(X)$. With the techniques of homology and linear algebra, we can assign to a simplicial complex $K_\epsilon := \mathrm{VR}_\epsilon(X)$ a set of groups, called the homology groups, $H_k(K_\epsilon)$. These groups describe topological features, the connected components and $k$-dimensional holes for $k \leq d$.

One thing that is special about $\mathrm{VR}_\epsilon(X)$ is that $\epsilon$ can be chosen such that the Vietoris-Rips is homotopy-equivalent to the manifold from which $X$ comes from; see Proposition 3.1 in Niyogi et al. (2008) for more details on how $\epsilon$ can be appropriately chosen to ensure the homotopy-equivalence property. In other words, descriptions of the topology of the right $\mathrm{VR}_\epsilon$ translate to the topology of the original manifold.

Everything described thus far is standard simplicial homology. However, when applied to noisy real data (represented as point clouds), small differences in the distance between points could imply a large difference in terms of the topology. To account for this, PH considers the Vietoris-Rips filtration, $\{K_\epsilon : \epsilon \geq 0\}$. As $\epsilon$ grows, the topology of $K_\epsilon$ changes as a result of increasing the radius of every ball in Equation (1). For example, connected components that appeared at scale $\epsilon = \epsilon_2$ (birth time) may merge into one component at scale $\epsilon_5 > \epsilon_2$ (death time). PH tracks those changes as $\epsilon$ increases from 0 to infinity, providing the birth and death time for topological features as well as their total number at each scale.

---

[1] We borrow the notation from Wheeler et al. (2021). The definition does not necessarily need a norm, but can be adjusted to work with a metric $d$.

We call the difference between the death time and the birth time of a topological feature the *persistence time* of the feature. Features with large persistence times tend to be the most topologically important Hensel et al. (2021). In this work, we only concern ourselves with the persistence times of the connected components (i.e., corresponding to $H_0$) which we get from the `ripser` Python library (Tralie et al., 2018). All such persistence times are positive numbers.

For the remainder of this manuscript and given a dataset $X \subset \mathbb{R}^d$, `ripser` will provide an array of persistence times, $p_i$, $i = 0, \ldots, M$, one for each connected component discovered. Because persistence times can grow with the diameter of $X$, $\text{diam}(X) = \max_{x,y} \|x-y\|$, we normalise them to $[0, 1]$ by dividing with the maximum persistence time (after removing the special persistence time $p_M = \infty$ at $\epsilon = \infty$). As a consequence, the distribution of the persistence times will be invariant to point-cloud scaling.

## 3 Class Separability

### 3.1 Baseline measures of separability

In this section, we briefly describe the measures we will use as proxies for separability.

**ROC-AUC**: An estimate of the area under the ROC curve for logistic regression models trained on labeled data.

**Accuracy**: An estimate of the (balanced) accuracy of logistic regression models trained on labeled data. This is calculated with the scikit-learn "balanced_accuracy" function.

**Thornton Index** (Greene, 2001): The Thornton index is the probability that a random data point's label is the same as that of each of the nearest neighbors. We estimate it by using every point's ten nearest neighbors.

ROC-AUC, accuracy, and the Thornton index require an additional set of labeled data, e.g., a validation set, to be estimated. For an unsupervised baseline for separability, we use the Calinski-Harabasz Index.

**Calinski-Harabasz Index** (CH) (Caliński and Harabasz, 1974): The CH is often used to measure clustering performance, e.g., of $k$-means. With $k$ clusters and $N$ data points, the index is proportional to $\text{SS}_B/\text{SS}_W$, where $\text{SS}_B$ is the between-cluster variance and $\text{SS}_W$ is the within-cluster variance. The larger the CH, the more well-defined the clusters are, and this is an unbounded metric as the clusters can always be more concentrated around their center. Because of this and to aid in comparisons, we normalize CH by its maximum value in every experiment.

We use CH by first applying k-means clustering[2] with $k = 10$ to the embeddings in order to assign them to clusters. Then we use the scikit-learn function "calinski_harabasz_score()" to estimate CH.

As we will discuss in the next section, we are interested in using information from $H_0$ to learn about how separability evolves during training. $H_0$ tracks the existence of connected components so a comparison against CH is instructive.

### 3.2 Using persistent homology to capture separability

The proposed method for estimating the class separability of a dataset $X$ is solely based on the persistence times of the 0-homology group of the data manifold, $H_0(X)$. We argue (and will explore experimentally) that tracking the evolution of the distribution of persistence times can provide information on how a classification model organizes its embedding space as well as whether we are getting diminishing returns by further training. As mentioned in Section 2, for every point cloud $X$ we compute the persistence times $\mathbf{p}$, remove the $+\infty$ persistence time, and then normalize the rest by the maximum persistence time. This procedure makes $\mathbf{p}$ invariant to point-cloud scaling.

We will often refer to "topologically-simple" point clouds, e.g., embedding vectors. This is a shorthand for "topologically-simple point clouds with respect to the $H_0$ homology". The simplest such point clouds

---

[2]We found that CH is not very sensitive to $k$ as long as $k$ is large enough.

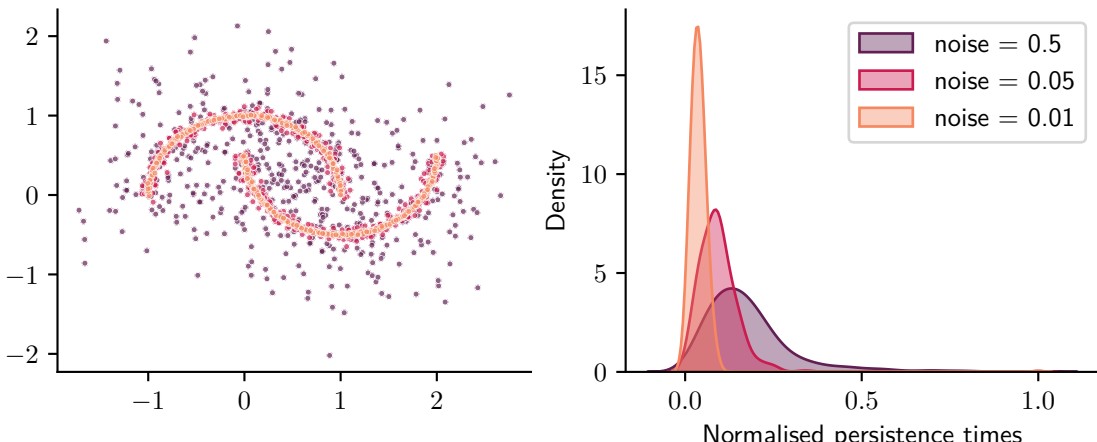

Figure 1: Evolution of the densities of persistence times for a moon dataset (sampled with "sklearn.datasets.make_moons") with different noise parameters. As noise decreases, the points in each cluster approach each other, making most (but not all!) persistence times small (see Equation (1)) and increasing the confidence that a small number of connected components exists.

are ones where connected components can be discerned with high confidence, e.g., two well-separated point clusters; this would correspond to only having a few large persistence times (corresponding to the largest connected components) and the vast majority of them close to the smallest persistence time. In the most complex cases, it is hard to discern more than one component and we would expect to see a larger spread of persistence times. An example of those behaviors can be seen in Figure 1.

To compare the information from the persistence times with the metrics from Section 3.1, we need to use an appropriate statistic. We first make two assumptions:

1. **Topologically-simpler embedding spaces result in easier classification problems**: This is a reasonable assumption that is supported by similar work referenced at the end of Section 1.1, e.g., the work of Griffin et al. (2023). Heuristically, if well-defined point clusters exist in the embedding space and each cluster contains mostly examples from one class, then the classification problem should be easier.

2. **During classification training, models tend to topologically simplify their embedding space**.

The inverse of Assumption 1 is not true in general and depends on how we define topological-simplicity. Consider the point clouds $X_1 = \{(1, i) : i \in \{1, \dots, 10\}\}$ and $X_0 = \{(0, i) : i \in \{1, \dots, 10\}\}$ and assign labels 1 and 0 to them respectively. Then the (finite) persistence times of the $H_0$ homology for the set $X_0 \cup X_1$ are all equal to 1 as all points have a nearest neighbor with that distance (refer to the definition of Vietoris-Rips, Equation 1). This indicates a single persistent component, yet the classification problem is linearly separable. This special case is not applicable to this work as we will study the evolution of embeddings during training.

Assumption 2 does not always hold (see Section 5), but is empirically supported for some cases (Naitzat et al., 2020). In practice, we can check it by tracking the evolution of the densities of the persistence times during training. We noted experimentally that we can also push the models towards this behavior during fine-tuning by normalizing the embeddings so that they have a norm equal to one (see Figure 3). We hypothesize that this effect is due to normalization and the max-margin convergence during optimization (Soudry et al., 2018; Ji and Telgarsky, 2019), essentially making the embeddings more concentrated as training progresses.

**Persistence score:** To summarize the behavior of the normalized persistence times and compare with the rest of the metrics, we need a statistic $S(\mathbf{p})$, where $\mathbf{p}$ denotes the set of normalized persistence times

of the point cloud $X$. Given Assumption 2, we would like $S$ to be bounded and to increase as most persistence times concentrate closer to $\min\{\mathbf{p}\}$, indicating more confidence that connected components exist (for intuition see Figure 1). A simple statistic of concentration is the expected squared deviation from $\min\{\mathbf{p}\}$, $\mathcal{E}[(\mathbf{p} - \min\{\mathbf{p}\})^2] = (\mathcal{E}[\mathbf{p}] - \min\{\mathbf{p}\})^2 + \mathcal{V}[\mathbf{p}]$, where $\mathcal{E}[\mathbf{p}]$ denotes the sample mean over the elements of $\mathbf{p}$ whereas $\mathcal{V}(\mathbf{p})$ denotes the sample variance.

We now study the possible behaviors of this statistic as it converges towards zero, so first we set $E[\mathbf{p}] := (\mathcal{E}[\mathbf{p}] - \min\{\mathbf{p}\})^2$ and $a[\mathbf{p}] := E[\mathbf{p}]/\mathcal{V}[\mathbf{p}]$. Then, we notice that $E[\mathbf{p}] + \mathcal{V}[\mathbf{p}]$ could converge to 0 while either (1) $a \to 1$ (both terms decay to zero at the same rate), or (2) $a \to 0$ (expectation term goes to zero faster), or finally (3) $a \to \infty$ (variance term goes to zero faster). While Case (1) and Case (2) suggest concentration close to $\min\{\mathbf{p}\}$, as the expectation term is controlled, Case (3) does not. For a fixed small $\delta > 0$, we would have $E[\mathbf{p}] + \mathcal{V}[\mathbf{p}] \le \delta$ with $E[\mathbf{p}] \approx \delta$, so we could have concentration away from $\min\{\mathbf{p}\}$. To decrease the influence Case (3) has on the statistic, we can increase the weight of the expectation term, finally defining $S_\beta(\mathbf{p}) := 1 - (\mathcal{E}[\mathbf{p}] - \min\{\mathbf{p}\})^\beta - \mathcal{V}(\mathbf{p})$, where $\beta \in (0, 2)$. $S_\beta$ retains the properties we discussed while placing more weight on the distance from $\min\{\mathbf{p}\}$. As our purpose is to summarize the behavior of $\mathbf{p}$, we use $S := S_1$ in this work. Because we are dealing with normalized persistence times, the statistic is invariant to changes in the diameter of the point cloud $X$.

By analyzing persistence times and the persistence score $S(\mathbf{p})$, our objective is to gain insights into the structure of the embedding space as models are trained. Given Assumption 1, we aim to assess the separability of embeddings from a topological perspective and better understand how embeddings evolve during the training process.

## 4 Experiments

This section includes experimental validation of the main scheme of the paper. Experiments involve a binary and a multi-class text example with pre-trained sentence transformers (ST).

Before each experiment, we split a text classification dataset into a training split, a tracking split, and a validation split. We then fine-tune the ST for text classification with cross-entropy loss on the train split. Once before training and then at the end of every epoch, we embed every element of the tracking split with the updated ST.

**How do we estimate the supervised metrics of separability?** We use cross-validation to estimate the model-dependent supervised metrics, balanced accuracy, and ROC-AUC, by training logistic-regression models on part of the embedded tracking set and computing the metrics on the remainder. We average over multiple splits to reduce the noise of model fitting on those estimates. Thornton's index, CH, and $S(\mathbf{p})$ are calculated once per epoch on the whole tracking set. After this, we can compare the behavior of the metrics per epoch.

To reduce the dependence on a particular train-tracking-validation split, we compute the scores over seven randomly-picked train-tracking-validation splits.

### 4.1 Binary-Class Text Classification

In this experiment, we start with ST and a set of binary classification datasets.

**Datasets**: We use the train splits of *SetFit/amazon_counterfactual* and the *SetFit/sst2* datasets from Hugging Face[3]. The amazon-counterfactual dataset contains text from English, German, and Japanese, as well as a label for whether the text is describing a counterfactual or not-counterfactual statement. Only 19% of the text describes counterfactuals. The train split of the sst2 dataset is close to being balanced, with 52.8% of the dataset having "positive" label.

---

[3]We only use the SetFit datasets (Tunstall et al., 2022) because of their column-name standardization, we do not use the SetFit paradigm in this work.

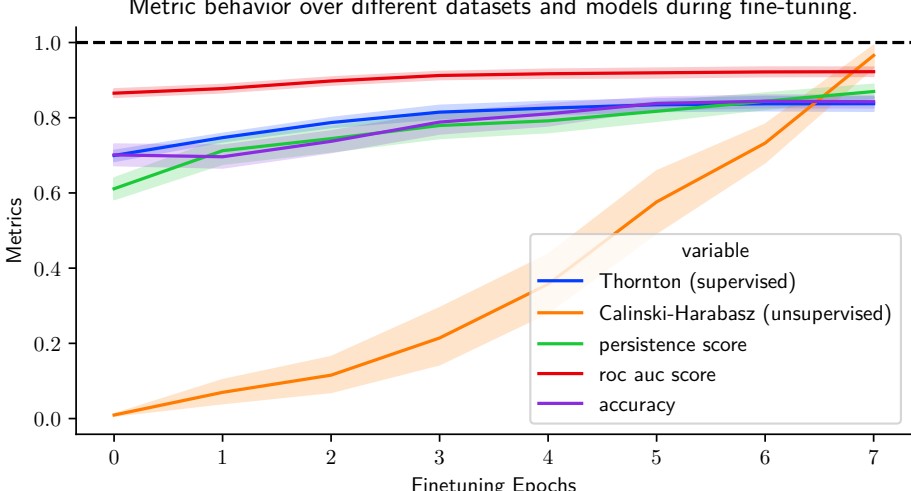

Figure 2: Separability metrics for the binary classification text example. Every line tracks the mean over all models (*all-MiniLM-L6-v2*, *paraphrase-TinyBERT-L6-v2*, and *sentence-transformers/paraphrase-albert-small-v2*) and datasets (the train splits of *SetFit/amazon_counterfactual* and *SetFit/sst2* datasets from Hugging Face). We use seven splits per dataset/model combination. Accuracy, Thornton, and our persistence-score exhibit similar behavior. Additionally, the increase in the persistence score indicates that the embeddings are organized into more well-defined components. Intervals are 95% confidence bands.

We split each set into a training set with 1000 examples and a tracking set with 1000 examples. At the end of every epoch, we embed the examples in the tracking set with each sentence transformer. Although we use those embeddings at the end of the run to gather information about the training process by computing how the various metrics of Section 3.1 changed, in practice those metrics can be computed during the training run.

**Models**: We consider three pre-trained sentence transformers available on Hugging Face and through the *sentence-transformers* Python library (Reimers and Gurevych, 2019)[4]. The models are sentence-transformers/*all-MiniLM-L6-v2* (MiniLM), sentence-transformers/*paraphrase-TinyBERT-L6-v2* (TinyBert), and *sentence-transformers/paraphrase-albert-small-v2* (Albert). MiniLM is a six-layer network outputting normalized 384-dimensional sentence embeddings, whereas TinyBert and Albert both output 768-dimensional unnormalized embeddings. All models produce sentence-embeddings via averaging over the token embeddings that are produced by the last layer. As per Section 3.2, we normalize the embeddings so that they all have a norm equal to one. For each of the above models, first before we start finetuning and then after every epoch, we create sentence embeddings for each element in the tracking set. Then we use the "ripser" Python library to compute the persistence times. This way we can inspect the change in the organization of the embeddings during finetuning.

**Finetuning Procedure:** We attach a randomly-initialized one-layer sigmoid head to construct a binary text classifier that outputs the probability of one of the classes. This is a classical configuration when fine-tuning a classification model, such as ALBERT (Lan et al., 2019). We fine-tune each model on the training splits of each dataset. We use the prodigy optimizer (Mishchenko and Defazio, 2023) with d_coef = 1e−1, cross-entropy loss, and batch size 32 for all models. Each training run lasts 7 epochs. As we fine-tune on relatively small datasets, we only train the last layer of each ST and the classifier head.

Figure 2 shows the evolution of the various metrics of separability on the tracking sets. The models never see the labels of these sets during training. As we train, the supervised metrics capture a small increase in the separability of the embeddings. The persistence metric exhibits similar behavior indicating this

---

[4]We use the "sentence-transformers" version "2.4.0" to load the models and encode text into sentence embedding vectors.

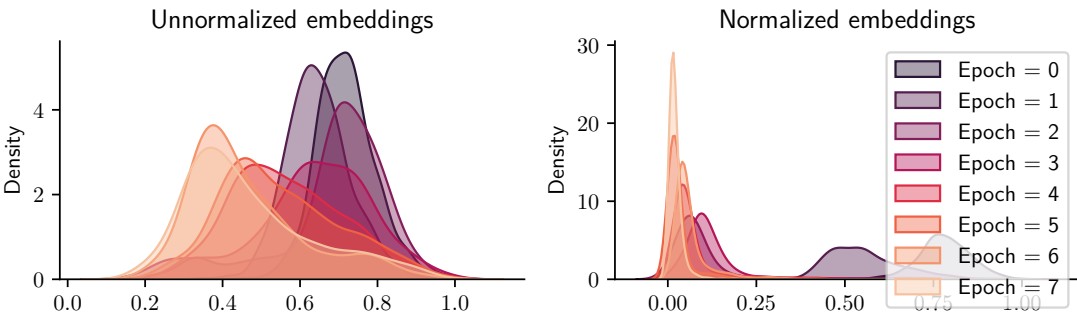

Figure 3: A comparison of the evolution of the normalized persistence times of the tracking set of the dataset *SetFit/amazon_counterfactual* for the *TinyBert* model during fine-tuning. On the right plot, TinyBert includes an additional normalization step that ensures the embeddings have a norm equal to one. Otherwise, the setup between the two plots is the same (same splits, optimizer configuration, weight initialization for the ST). The point cloud is organized during fine-tuning so that the connected components are well-separated on the unit ball (compare with Figure 1). On the left plot, no normalization is taking place and there are more persistence times close to one, indicating a different organization of the embedding space that nevertheless also gets an acceptable balanced accuracy (0.79 vs 0.88 for the normalized case in this example). We remind here that the persistence scores are normalized so they are invariant to point-cloud scaling.

slow increase as well. However, because of the topological motivation behind the persistence score, this convergence to larger values also implies that we are approaching a topologically simpler solution to the classification problem. Finally, the CH metric keeps on increasing throughout fine-tuning, showing the qualitative difference between the notion of separability it captures versus the one from the persistence score. In Figure 3 we can see the distribution of normalized persistence times for the case of normalized and unnormalized embeddings on the SetFit/amazon_counterfactual data and the TinyBert model. We note that the normalised case shows behavior that is consistent with our previous Assumption 2.

## 4.2 Multi-Class Text Classification

In this experiment, we consider the behavior of pre-trained sentence transformers during fine-tuning for multi-class classification. The setup is similar to the binary class experiment in Section 4.1.

**Datasets**: We use the train split of two datasets: *SetFit/emotion* and *financial_phrasebank*, both of which can be found on Hugging Face.

The dataset *SetFit/Emotion* contains six classes (with approximate corresponding appearance in the set): "joy" (33.5%), "sadness" (29.1%), "anger" (13.5%), "fear" (12.1%), "love" (8.1%), and "surprise" (3.6%). The dataset *financial_phrasebank* contains three classes: "neutral" (62%), "positive" (25.7%), and "negative" (12.2%). Both datasets contain English text. The financial_phrasebank dataset variant we use is labeled according to a 75% agreement between annotators.

**Model**: We use the models and the training and tracking setup from Section 4.1, except now the head of each ST leads to a softmax with the probability of each class.

Thornton's index and accuracy continue to have very similar behavior in Figure 4 and Figure 5. Our persistence score also mimics those behaviors and, more importantly, shows the diminishing returns after a few epochs of training. In both cases CH keeps on increasing throughout the epochs, indicating that the clusters keep on becoming better defined.

Figure 6 is similar to the binary classification example we saw in Figure 3. Here too we see that when the embeddings are normalized, the embedding space gets more well-defined connected components in com-

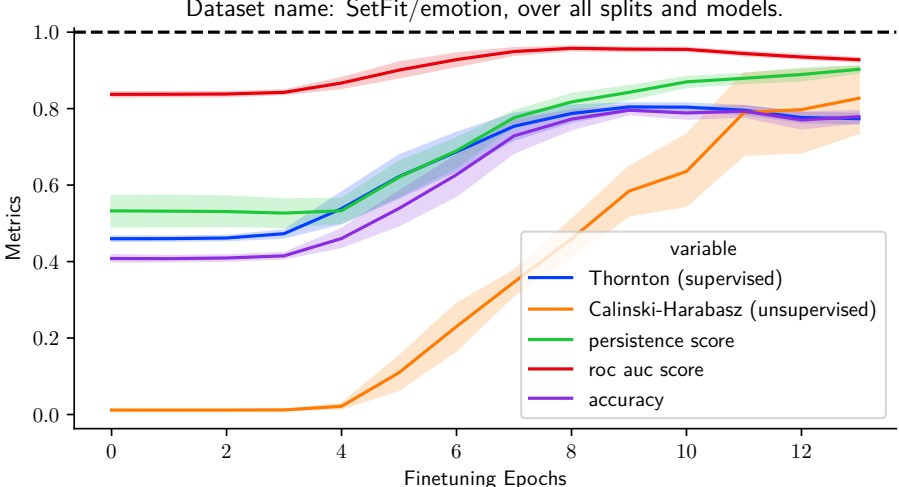

Figure 4: The behavior of the various separability metrics on the embeddings as epochs progress. Each line includes a 95% confidence interval and is composed of the corresponding metric over seven splits and the three models: *all-MiniLM-L6-v2*, *paraphrase-TinyBERT-L6-v2*, and *sentence-transformers/paraphrase-albert-small-v2*. The persistence score mimics the behavior of the supervised metrics, including the upward arc starting at epoch 4 and subsequent slowdown. The CH score continues to increase linearly throughout the epochs after epoch 4.

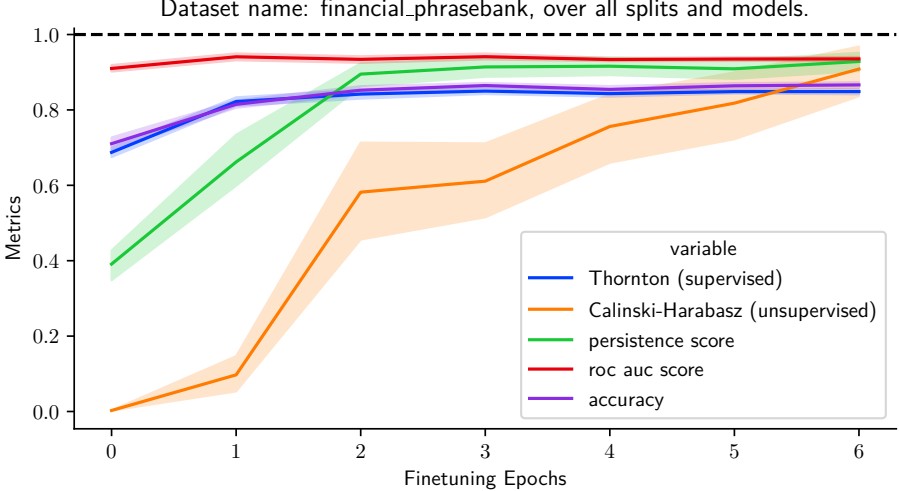

Figure 5: Another example of behavior of metrics through training; the setup is the same as for Figure 4. In this example, we notice a saturation in terms of separability after epoch 1 for the supervised metrics and epoch 2 for the persistence score. The CH score never truly saturates indicating that the clusters continue to become more separated in the embedding space, yet any further difference does not make such a difference to the rest of the metrics.

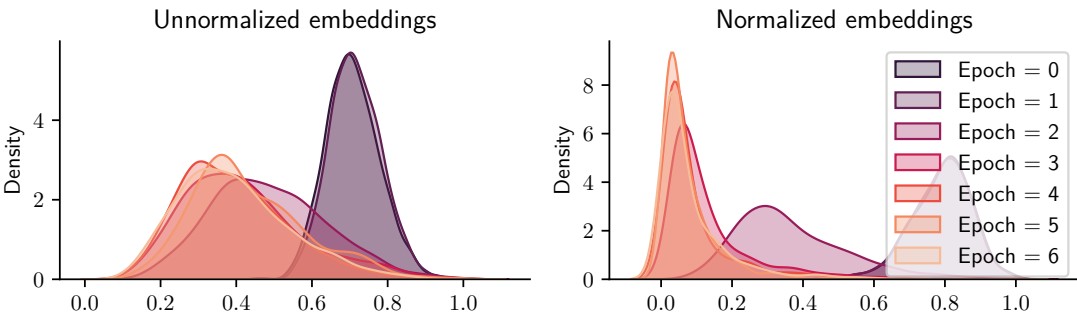

Densities of $H_0$ persistence times

Figure 6: Evolution of the densities of the normalised persistence times of $H_0$ for the multi-class *Set-Fit/Emotion* dataset and the *TinyBert* model. The setup is similar to Figure 3. We can see that the behavior is similar to the binary case; the ST with normalized embeddings jumps through distinct states and eventually reduces most persistence times, whereas the persistence time distribution in the unnormalized case does not change that much.

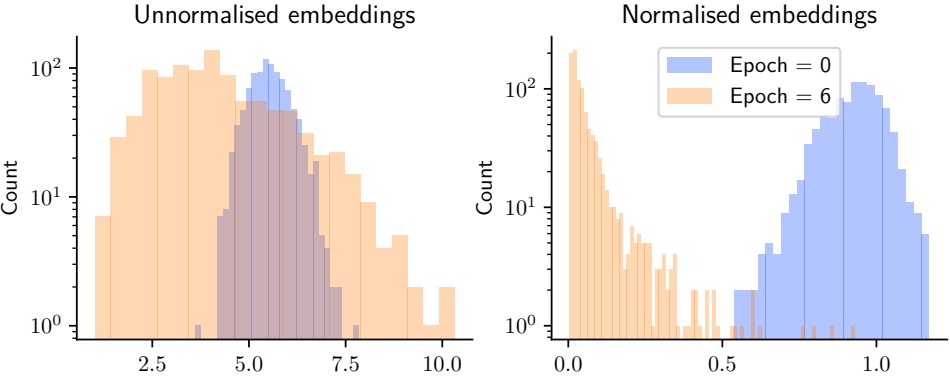

Figure 7: This is the case described in Figure 6 (dataset: SetFit/Emotion, model: TinyBert), but we remove the normalization of the persistence scores; now the persistence scores are also affected by the diameter of the embedding set. We have switched the y-axis to log scale to make the tails of each distribution more visible. In the left plot, we can see that the distances between some of the connected components become larger from epoch 0 to epoch 6, indicating that the model is partially using the magnitude of the embeddings to separate the classes. We expect that, after a certain point, this increase in distance will not make a difference in terms of separability; see Figure 5 and Figure 4. In the normalized case on the right, we see the expected collapse of most of the persistence times.

parison to the unnormalized case as finetuning progresses. To provide further insight in this behavior, in Figure 7 we look at the distribution of the original persistence scores, i.e., without any normalization. In this case, the persistence scores depend also on the diameter of the embedding set and we can see that in the unnormalized case the embedding model is also adjusting the magnitude of the embeddings to separate the classes.

## 5 Discussion

**Computational complexity of $H_0$ computations**: The computational bottleneck of our study is the computation of the persistent homology of the point clouds. Methods and algorithms for computing persistent homology is a rapidly evolving area, where new algorithms and software implementations are being

updated and released at a rapid pace (Otter et al., 2017). In this paper, we use the `ripser` library for computing the persistent homology of datasets (Bauer, 2021). Vietoris-Rips complex calculations have a runtime complexity of $O(2^n)$, where $n$ is the dataset size, but constant with data dimension, which makes this method preferable for high-dimensional manifolds (Somasundaram et al., 2021). Variants exist that make use of data-reduction techniques to produce approximations of the persistence times for a fraction of the computational cost; a comparison of different methods can be found in the work of Malott et al. (2020). Apart from their computation-saving advantages, we also think they are interesting because such methods can focus on "higher-scale" structure in the point cloud, e.g., between connected components that contain a minimum number of points, while decreasing the effect of lower-scale structure between points. We plan to explore the application of those methods in future work.

**Selection of the summary statistic**: While we hope that we have shown the usefulness of tracking the distribution of the persistence times, we do not claim that the statistic $S(\mathbf{p})$ is optimal when it comes to summarizing persistence time behavior. Its usefulness depends on the assumptions we make in Section 3. Our focus was in having a way to compare the behavior of the persistence times alongside common separability statistics. In future work we would like to study the sensitivity of the results to this statistic on other tasks, such as text generation, and better formalize its selection.

**Behavior of persistence times during training:** While the persistence times of the embedding manifold do not use label information explicitly for their calculation, their behavior depends on the training labels, the model architecture, and the training objective. As we have seen, normalizing the embeddings is one way to control the behavior of the persistence times. With the constraint of normalization, the classifier has to use angles to organize the embeddings for the best-possible separability which leads to the clustering we see. Further experiments are required to understand the dynamics of persistence times in the case of unnormalized embeddings.

## 6   Conclusion

This paper proposes using information from the topological characteristics of the data manifold to look at the training process from a different angle. We showed experimental evidence that the information of the $H_0$ homology group of the embeddings can give valuable information on where we are in the training process: is separability still improving or have we reached a plateau? To be able to do this, we need to expect a certain behavior from the persistence times and we showed evidence that most of those collapse during training when the embeddings are normalized. When this happens, our summarising statistic shows similar behavior with the supervised metrics, but also gives us information about the shape of the embedding space (in terms of concentration of the connected components).

Potential topics for future work involve:

- Formalizing the selection of the summary statistic and how that depends on the architecture of the model we train.

- Expanding the analysis to the densities of persistence times of higher homology groups $H_n$ of data manifold and their relations to class-separability of the dataset.

- Expanding the study to tasks outside classification: regression, text generation, and model probing.

**Disclaimer**

offer, or solicitation for the purchase or sale of any security, financial instrument, financial product or service, or to be used in any way for evaluating the merits of participating in any transaction.

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
