# OpenReview forum: "Estimating class separability of text embeddings with persistent homology."
_TMLR — Accepted by TMLR_

### Review · Reviewer_jAxw · 2024-03-22

**Summary Of Contributions:**

This work develops a method for estimating class separability without labels using the tools of persistent homology.

Specifically, the work:

* Argues that a simpler embedding space results in easier classification problems
* Measures the persistence time of the $H_0$ group
* Measures $S(p)$, the concentration around the minimum persistence time, as a summary of the simplicity of the embedding space
* Observe that the measure $S(p)$ behaves consistently with supervised measures like Thornton (supervised) and accuracy as a function of training.

Additionally, the authors find that in the case of L2-normalized embeddings, models push the distribution of persistence times to be concentrated around 0 during training, whereas the pattern is less clear without L2-normalization.

**Audience:**

Yes

**Broader Impact Concerns:**

No broader impact concerns.

**Claims And Evidence:**

Yes

**Requested Changes:**

# Required (critical)

* Restructure paper to colocate figures with in-text references.
* Explanation in-text for the choice of $S(p)$, addressing the scaling behavior of the terms under $p\rightarrow cp$, explaining why summing the expectation, min and variance is a reasonable thing to do.
* Explain what space the persistent homology is acting upon in Section 4 for each model.
* Improve presentation of figures as described above.
* (Page 3, first line) Typo on quotes, change persistence" → persistence''
* (Page 3, paragraph 1) $H_0$ is used without definition, add definition before this point.


# Recommended for strengthening but not critical

* (Section 1.2) “Major contributions” → “Primary contributions”
* (Section 1.2, first bullet) “We leverage” → “We use”
* (Section 1.2, second bullet) State the conclusion of the analysis here, not just the existence of analysis
* (Section 1.2, second bullet) State the conclusion of the analysis here, not just the existence of analysis
* (Figure 2, page 7) Mention which datasets are used in the caption.
* (Page 4, under equation 1) I think this is a typo but not 100%, if it is, please change. $\textrm{VR}_{\infty}=$ {{ $ x_1,\ldots,x_N $ }}, i.e. the set containing a single sigma which is the set of all points in X.
* (Page 4, penultimate paragraph) provide citation for “Features with large persistence times tend to be the most topologically important”.
* (Page 6) It would be useful to have a \paragraph{Persistence score $S(p)$} to call out the definition of the score you will use in Section 4.
* (Page 6) Add a simple clustering figure showing that your $S(p)$ gives the correct ordering in the right scenario. I understand this is the purpose of Figure 1. Since most of the text discusses well-formed clusters, it might be useful to include e.g. 1) random box uniform noise (low separability) and 2) two tight clusters (high separability). Compute $S(p)$ for each case and have them as figure subtitles.
* (page 8) Add a \paragraph{Finetuning procedure} to call out where you discuss finetuning procedure (distinct from models).
* (page 10, Figure 6) Change caption end “that dramatically” → “as much”
* Remove instances of “etc.” throughout the work and complete the sentences. E.g. Conclusion final point “regression, text generation, etc.” → “for example, regression and text generation”.

**Strengths And Weaknesses:**

# Strengths

Measuring class separability without labels is of interest to many ML communities. This is true for self-supervised learning, where ideally no labels are used (leading to methods like RankMe https://arxiv.org/abs/2210.02885 and LiDAR https://arxiv.org/abs/2312.04000). Labels may also be ambiguous, change with time, or be few in number.

The work is mostly self-contained, providing a useful guide through the concepts of persistent homology needed for understanding the proposed method and results. Additional references are provided for further technical details.

The work provides a good statistical analysis of the claims, using thorough cross-validation over many splits, as well as reporting confidence intervals where appropriate.

# Weaknesses

The layout of the results is challenging to follow, e.g.:

* Figure 1 (page 7) is only mentioned in the caption of Figure 3 (page 8).
* Figure 2 (page 7) is only mentioned on page 9
* Figure 3 (page 8) is only mentioned on page 6 and in the caption of Figure 6 (page 10)

Figures and tables should be anchored by in-text references, and should occur as close as possible (preferably on the same page) as those references.

The paper is unclear which exact object is having the persistent homology being tracked. In Section 3.2 the authors discuss tracking $H_0(X)$ where $X$ is the dataset, as well as under Section 1.2. In the context of their experiments in Section 4 this cannot be the case as $X$ does not change with time. I think the authors mean to track some specific embedding layer(s) of the models they are looking at. For example, the CLS token, or the max or mean pool of the last L layers of the model in equation. The paper would be much improved if the authors included a description of which objects their metric is tracking in Section 4.

I find the chosen definition of $S(p)$ unusual. I agree that the statistic obtains the correct values for $S(p)=0$. My specific question regards adding expectation, min and variance operators together. If we scale the persistence times by a constant, then the current formulation of $S(p)$ places more weight on the $\textrm{Var}(p)$ term, since $\textrm{Var}(cp)=c^2\textrm{Var}(p)$ for some constant c, whereas $\mathbb E(cp)=c\mathbb E(p)$ and similarly for min. This feels undesirable. Wouldn’t it make more sense to compute instead $S(p)=1-(\mathbb E(p) - \textrm{min}(p) + \textrm{std}(p))$ which would have a well-defined scaling behavior. Alternatively, why not compute directly the motivated $1-\mathbb E[(p-\textrm{min}(p))^2]$ (potentially with a sqrt)?

The presentation of the figures can be improved.  Figures should be vector graphics where possible. Text size on all figures should more closely match document text size. Captions for variable names should not contain underscores unless they are important. x and y axes should be labeled, and be capitalized if appropriate (e.g. epoch on Figure 2 should be Training Epochs or Finetuning Epochs). Other suggested recablings include Figure 3 x axis should be labelled $S(p)$ and y axis should be labelled Density $S(p)$.

---

> ### Author Response · Authors · 2024-04-04
> **Response to Reviewer jAxw**
>
> We would like to thank the reviewer for the in-depth engagement with our work. We first discuss the changes in the manuscript and then go over particular questions.
>
> # From the *"Recommended for strengthening but not critical"* section
>
> We added 99% of the suggested changes. The one change we didn't implement exactly as written was:
>
> > Add a simple clustering figure showing that your  gives the correct ordering in the right scenario. I understand this is the purpose of Figure 1. Since most of the text discusses well-formed clusters, it might be useful to include e.g. 1) random box uniform noise (low separability) and 2) two tight clusters (high separability). Compute for each case and have them as figure subtitles.
>
> Instead we refreshed Figure 1. but added the case of noise=0.5 in which the original connected components are much less visible.
>
> # From the *Required (critical)* section
>
> We made changes as follows:
>
> - "Restructure paper to colocate figures with in-text references."  → Done, should be better now.
> - "Explanation in-text for the choice of , addressing the scaling behavior of the terms under , explaining why summing the expectation, min and variance is a reasonable thing to do."  → Added a discussion on this in the "Persistence score" paragraph, Section 3.2. We start from the better motivated squared deviation and work towards our final statistic instead of the other way around.
>
> - "Explain what space the persistent homology is acting upon in Section 4 for each model."  → Done, added text in the "Models" paragraph in Section 4.1.
>
> - "Improve presentation of figures as described above."  → Done, all plots are now vector graphics and the font used should be closer in size and style with the manuscript.
>
> - "(Page 3, first line) Typo on quotes, change persistence" → persistence''  → Fixed.
> - "(Page 3, paragraph 1)  is used without definition, add definition before this point."  →  Done, added text close to the appearance of $H_0$ to clarify what that object is.
>
> # Questions
>
> > The paper is unclear which exact object is having the persistent homology being tracked. In Section 3.2 the authors discuss tracking
>  where is the dataset, as well as under Section 1.2. In the context of their experiments in Section 4 this cannot be the case as
>  does not change with time. I think the authors mean to track some specific embedding layer(s) of the models they are looking at. For example, the CLS token, or the max or mean pool of the last L layers of the model in equation. The paper would be much improved if the authors included a description of which objects their metric is tracking in Section 4.
>
> We added a comment in Section 4.1, Paragraph "Models" to make it more clear that we are embedding a set of points per epoch and then computing their persistence times. The text is:
>
> "For each of the above models, first before we start finetuning and then after every epoch, we create sentence embeddings for each element in the tracking set. Then we use the “ripser” Python library to compute the persistence times. This way we can inspect the change in the organization of the embeddings during finetuning."
>
> >I find the chosen definition of $S(p)$ unusual ...
>
> Thank you the question. To better motivate the selection of the statistic, we wrote the "Persistence Score" paragraph in Section 3.2 from the perspective of starting from $E[(p-\min\{p\})^2]$ and working towards the chosen $S(p)$. The core argument is that $E[(p-\min\{p\})^2]$ splits in two terms that essentially put similar weight to the variance of the persistence times and the distance from $\min p$. This is not something we want as $E[(p-\min\{p\})^2]$ can be almost equal to $(E[p]-\min\{p\})^2$ and $V[p]\approx 0$ (in which case the statistic is small without concentration necessarily being stronger). In other words, the expectation term should be more important than the variance term. There is some leeway there to use different weights for the expectation term, but as we just want to summarise the behavior of the persistence times, we settled for the exponent being equal to one (see the new paragraph "Persistence Score" in text).
>
> Also, because of the difference in importance of the terms, using std instead of var would require a subsequent adjustment to the expectation term.
>
> >  If we scale the persistence times by a constant, then the current formulation of places more weight on the [variance] term ...
>
> This is an issue that is not just theoretical as the scaling of the point cloud scales the persistence times too. However the statistic deals with the normalised persistence times, so scaling factors are removed, i.e., the statistic is invariant to such transformations.

---

> > ### Comment · Reviewer_jAxw · 2024-04-22
> > **Reply to response**
> >
> > I thank the authors for their detailed response and updates to the paper. The readability of the paper is much improved, and the procedure the authors propose and perform is clearer, and overall more reproducible.
> >
> > Addressing the two points that may still be in question:
> >
> > > Instead we refreshed Figure 1. but added the case of noise=0.5 in which the original connected components are much less visible.
> >
> > I am happy with the modification of Figure 1 as it succeeds on conveying the required information.
> >
> > > To better motivate the selection of the statistic, we wrote the "Persistence Score" paragraph in Section 3.2
> > Thank you for performing this rewrite. The reordering allows the ideas to flow logically and demonstrates why on its own, expected squared deviation is insufficient.
> >
> > One suggestion I have that might make $\delta$ argument clearer would be to parameterize $E[p]$ and $\mathcal V[p]$ in terms of a parameter, e.g. $\alpha$, and use a presentation like $\alpha = E[p]/\mathcal V[p]$, and construct the limits of $E[p] + \mathcal V[p] \leq \delta$ as $\delta\rightarrow 0$ for $\alpha\rightarrow\infty$,  $\alpha\rightarrow0$ and $\alpha\approx 1$, which correspond to the three situations made in the paragraph. Independently, it may be useful to number each of these cases i), ii) and iii) in the paragraph to call them out, or use bullets.
> >
> > I still am not fully convinced by the naturalness of this statistic, as the highlighted issue appears to stem from absence of statistical (not feature) normalization, which implies one should use variance normalized quantities or (min-var equivalents). However, it is a statistic that has properties the authors desire for their analysis, and its origin is now clearly explained for a reader. Given this choice of statistic, the results of the paper do follow.
> >
> > Given the discussion provided by the authors, the chosen statistic is not problematic for me in terms of paper acceptance. However, I do think that investigating how sensitive the proposed type of analysis is to the choice of concentration statistic would be an important future contribution as it is unclear to me how robust the proposed analysis is to this choice. I also think it would be worth highlighting towards the end of the paper as a limitation: the choice of concentration may or may not be important, is an addition to the persistent homology toolbox, and overall conclusions may be sensitive to this choice.

---

> > > ### Author Response · Authors · 2024-04-30
> > > **Reply to Reviewer jAxw**
> > >
> > > Once again, thank you for the engagement!
> > >
> > > > One suggestion I have that might make $\delta$ argument clearer ...
> > >
> > > We incorporated this change in the manuscript.
> > >
> > > > I still am not fully convinced by the naturalness of this statistic, as the highlighted issue appears to stem from absence of statistical (not feature) normalization, which implies one should use variance normalized quantities or (min-var equivalents).
> > >
> > > >However, I do think that investigating how sensitive the proposed type of analysis is to the choice of concentration statistic would be an important future contribution as it is unclear to me how robust the proposed analysis is to this choice. I also think it would be worth highlighting towards the end of the paper as a limitation: the choice of concentration may or may not be important, is an addition to the persistent homology toolbox, and overall conclusions may be sensitive to this choice.
> > >
> > > We note the comments and we will be looking into this for future work. We have added a small piece of text in "Discussion", paragraph "Selection of the summary statistic" to make this a bit more clear.

---

### Review · Reviewer_oc5T · 2024-03-25

**Summary Of Contributions:**

This paper proposes an unsupervised topological method to estimate the class separability of text data during model training. By tracking the evolution of the 0-homology group of the embedding manifold using persistent homology, the authors argue that monitoring changes in the persistence time distribution provides insights into how well the embedding space separates classes. Experiments on text classification tasks demonstrate alignment between the proposed persistence-based score and supervised separability metrics.

**Audience:**

Yes

**Claims And Evidence:**

Yes

**Requested Changes:**

Exploring alternative summary statistics or providing a principled way to select S(p) could strengthen the method. Incorporating higher homology group information may enhance the approach's capabilities. Discussing computational complexity would address practical scalability concerns.

Expanding experiments beyond text classification could further validate the method's generality. More insights into interpreting persistence times and their relationship with separability could deepen the theoretical understanding.

**Strengths And Weaknesses:**

The paper proposes a novel unsupervised approach to assessing separability without labeled data. The promising empirical results showing alignment with supervised metrics are encouraging. The analysis of how embedding normalization influences persistence times and separability is insightful.

The choice of the summary statistic S(p) appears heuristic and may not generalize optimally. Only using 0-homology information limits richer topological insights from higher groups. The computational complexity of persistent homology is not discussed, which could hinder scalability.

---

> ### Author Response · Authors · 2024-04-04
> **Response to Reviewer oc5T**
>
> >Exploring alternative summary statistics or providing a principled way to select S(p) could strengthen the method.
>
> We thank the reviewer for this comment; we wrote a new paragraph "persistence score" in Section  3.2 to expand on the rationale of the statistic. As we expand in the discussion section (paragraph "Selection of the summary statistic"), this statistic only has a few good properties and a simple interpretation, and is meant only as a way to compare the information from $H_0$ with that of the other supervised and unsupervised statistics.
>
> > Incorporating higher homology group information may enhance the approach's capabilities.
>
> We agree with the reviewer that higher-order homology information is not used in this work and could improve the approximations. We focus on $H_0$ as
>
> 1.	We believe the persistence times associated with $H_0$ are more intuitive to practitioners unfamiliar with persistent homology as they relate to the connected components of the data manifold whereas higher-order $H$'s deal with k-dimensional holes.
> 2.	We wanted to assess whether information from $H_0$ is sufficient for the purposes of separability. Empirical evidence in the manuscript shows that is the case when the assumptions we introduce hold.
>
> However, we are interested in expanding to higher-homology groups for follow-up work on other tasks. We have this as future work in the “Conclusion” section.
>
> >Expanding experiments beyond text classification could further validate the method's generality.
>
> We have jotted that down as future work; we think there is value in communicating the simple results in the current manuscript for classification before expanding to cases where the benefit of having well-defined connected components, e.g., in regression, is not clear (or even required). Also, cases like regression and text generation may benefit from higher-order topological information which we don't cover in this manuscript.
>
> > Discussing computational complexity would address practical scalability concerns.
>
> We already have a brief discussion on computational complexity in the paragraph titled “Computational complexity of H0 computations” in Section 5, touching on the complexity of the "ripser" package calculations as well as references to efforts to alleviate this cost by using, e.g., clustering methods to reduce the amount of data used for computation. The computations in the ripser package are the main bottleneck of our approach which is why we don't have a separate section for computational complexity.
>
> >More insights into interpreting persistence times and their relationship with separability could deepen the theoretical understanding.
>
> We think the alternative way of looking at distributions of persistence times (see Figure 1. , 3., 6., and 7.) instead of just through the summary statistic provides some insights not just of separability, but of how the embedding space is organized during fine-tuning to facilitate separability (and how that depends on choices like embedding normalisation).
>
>  Inspired by this comment, we also added some additional text in the manuscript to highlight the comparisons between densities of persistence times (new text is at the end of Section 4.2, "Figure 6 is similar to the binary classification example ..."). Most of that information can also be found in the captions of the density plots / histograms.

---

> > ### Comment · Reviewer_oc5T · 2024-04-21
> > **Reply to Response**
> >
> > Thank you for the detailed response addressing the points raised in my review. I appreciate the clarifications and additional context provided. Overall, the authors' response addresses the concerns raised in the review and provides further context and clarifications. The openness to exploring alternative summary statistics, incorporating higher homology groups, and expanding to other domains in future work is encouraging. The additional discussions and visualizations enhance the theoretical understanding of the proposed approach.

---

> > > ### Author Response · Authors · 2024-04-30
> > > **Thank you!**
> > >
> > > We appreciate the comments!

---

### Review · Reviewer_4hwr · 2024-03-26

**Summary Of Contributions:**

The paper introduces a novel approach for analyzing the training process of text embeddings from a topological perspective.

**Audience:**

Yes

**Claims And Evidence:**

No

**Requested Changes:**

It is critical to conduct experiments to evaluate the proposed method's impact on model performance in downstream tasks such as classification or regression. This evaluation is essential for demonstrating the practical utility and effectiveness of the approach. Additionally, it would strengthen the paper's contribution and relevance to the field.

**Strengths And Weaknesses:**

The paper presents a novel method for tracking the training process of text embeddings without requiring additional labels. This innovative approach leverages information from the H0 homology group of embeddings to assess the progression of training, providing valuable insights into whether separability improves or plateaus.

One notable weakness is the absence of experimental evidence demonstrating how the proposed method contributes to achieving better performance or guiding model training effectively.

---

> ### Author Response · Authors · 2024-04-04
> **Response to Reviewer 4hwr**
>
> > One notable weakness is the absence of experimental evidence demonstrating how the proposed method contributes to achieving better performance or guiding model training effectively.
>
> We thank the reviewer for this comment as it shows that we need to highlight a bit more what can be done with the proposed metric. As the reviewer correctly noted, we can get an indication of when separability plateaus by tracking the behaviour of the $H_0$.
>
> Evidence that this is a meaningful comparison is presented in Figures 2., 4., and 5., which deal with multiple binary and multi-class classification realistic datasets and multiple sentence-transformer models.  In those examples, we can see that the information from $H_0$ and the statistic can capture not just the diminishing returns from training further, but also topological aspects of the final solution, e.g., whether the sentence-transformer model organizes the embeddings in terms of well-defined components or not. Because we utilise normalised persistence times, downstream statistics of the persistence times are not influence by the diameter of the embedding set. This means we ignore superfluous information that does not make much difference in terms of separability, e.g., in Figures 2., 4., and 5., compare the behaviour of the CH statistic with the rest of the metrics. We think those properties have value to practitioners during model fine-tuning / training (in addition to the study of $H_0$ not requiring additional validation labels).

---

### Author Response · Authors · 2024-04-04
**Revision of manuscript due to review.**

We would like to thank the reviewers for their time and efforts reviewing our paper. The feedback and comments they provided have been insightful and helpful to improve our submission.

Based on these comments we made several edits (marked in blue) in the paper.

The changes are as follows:

- Fixed typos and references (as indicated in the reviews).
- Updated figures to vector graphics and fixed fonts.
- Moved figures closer to their text references. Added text references for figures that didn't have them.
- New paragraph for the persistence score (as indicated by questions from Reviewer oc5T and Reviewer  jAxw).
- Additional text to clarify what is being tracked and when (Section 4.1, paragraph titled "Models", towards the end) as requested by Reviewer jAxw.

We will be responding to individual questions separately.

---

### Decision · Action_Editor_T6k5 · 2024-05-07

**Recommendation:** Accept with minor revision

**Comment:**

The paper presents an unsupervised, topology-motivated method for estimating the class separability of text datasets. Most reviewers believed that the proposed method was solid, novel, and effective. However, the reviewers also noted that the choice of the summary statistic S(p) appeared heuristic and the lack of complexity and scalability analyses. The AE believes that the paper has enough merits to warrant acceptance, and encourages the authors to address these concerns in the revision.

**Audience:**

Yes

**Claims And Evidence:**

Yes